# FlowQA: Grasping Flow in History for Conversational Machine Comprehension

**Hsin-Yuan Huang**[*]
California Institute of Technology
hsinyuan@caltech.edu

**Eunsol Choi**
University of Washington
eunsol@cs.washington.edu

**Wen-tau Yih**
Allen Institute for Artificial Intelligence
scottyih@allenai.org

## Abstract

Conversational machine comprehension requires the understanding of the conversation history, such as previous question/answer pairs, the document context and the current question. To enable traditional, single-turn models to encode the history comprehensively, we introduce Flow, a mechanism that can incorporate intermediate representations generated during the process of answering previous questions, through an alternating parallel processing structure. Compared to approaches that concatenate previous questions/answers as input, Flow integrates the latent semantics of the conversation history more deeply. Our model, FlowQA, shows superior performance on two recently proposed conversational challenges (+7.2% $F_1$ on CoQA and +4.0% on QuAC). The effectiveness of Flow also shows in other tasks. By reducing sequential instruction understanding to conversational machine comprehension, FlowQA outperforms the best models on all three domains in SCONE, with +1.8% to +4.4% improvement in accuracy.

## 1 Introduction

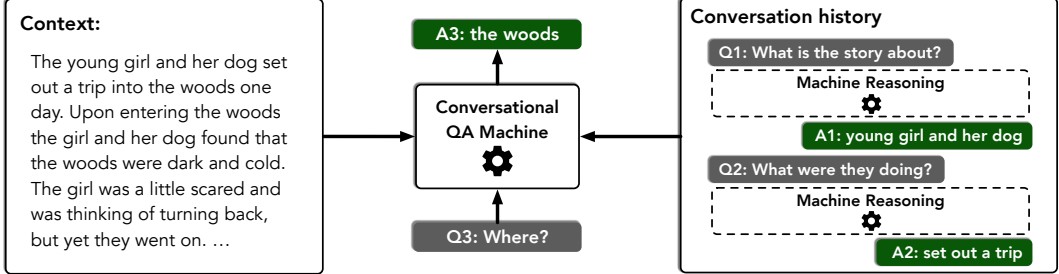

Figure 1: An illustration of conversational machine comprehension with an example from the Conversational Question Answering Challenge dataset (CoQA).

Humans seek information in a *conversational* manner, by asking follow-up questions for additional information based on what they have already learned. Recently proposed conversational machine comprehension (MC) datasets (Reddy et al., 2019; Choi et al., 2018) aim to enable models to assist in such information seeking dialogs. They consist of a sequence of question/answer pairs where questions can only be understood along with the conversation history. Figure 1 illustrates this new challenge. Existing approaches take a single-turn MC model and augment the current question and context with the previous questions and answers (Choi et al., 2018; Reddy et al., 2019). However, this offers only a partial solution, ignoring previous *reasoning*[1] processes performed by the model.

---

[*]Work done during internship at Allen Institute for Artificial Intelligence.
[1]We use "reasoning" to refer to the model process of finding the answer.

We present FLOWQA, a model designed for conversational machine comprehension. FLOWQA consists of two main components: a base neural model for single-turn MC and a FLOW mechanism that *encodes the conversation history*. Instead of using the shallow history, i.e., previous questions and answers, we feed the model with the entire hidden representations generated during the process of answering *previous* questions. These hidden representations potentially capture related information, such as phrases and facts in the context, for answering the previous questions, and hence provide additional clues on what the current conversation is revolving around. This FLOW mechanism is also remarkably effective at tracking the world states for sequential instruction understanding (Long et al., 2016): after mapping world states as context and instructions as questions, FLOWQA can interpret a sequence of inter-connected instructions and generate corresponding world state changes as answers. The FLOW mechanism can be viewed as stacking single-turn QA models along the dialog progression (i.e., the question turns) and building information flow along the dialog. This information transfer happens for *each context word*, allowing rich information in the reasoning process to flow. The design is analogous to recurrent neural networks, where each single update unit is now an entire question answering process. Because there are two recurrent structures in our modeling, one in the context for each question and the other in the conversation progression, a naive implementation leads to a highly unparallelizable structure. To handle this issue, we propose an alternating parallel processing structure, which alternates between sequentially processing one dimension in parallel of the other dimension, and thus speeds up training significantly.

FLOWQA achieves strong empirical results on conversational machine comprehension tasks, and improves the state of the art on various datasets (from 67.8% to 75.0% on CoQA and 60.1% to 64.1% on QuAC). While designed for conversational machine comprehension, FLOWQA also shows superior performance on a seemingly different task – understanding a sequence of natural language instructions (framed previously as a sequential semantic parsing problem). When tested on SCONE (Long et al., 2016), FLOWQA outperforms all existing systems in three domains, resulting in a range of accuracy improvement from +1.8% to +4.4%. Our code can be found in `https://github.com/momohuang/FlowQA`.

## 2 BACKGROUND: MACHINE COMPREHENSION

In this section, we introduce the task formulations of machine comprehension in both single-turn and conversational settings, and discuss the main ideas of state-of-the-art MC models.

### 2.1 TASK FORMULATION

Given an evidence document (context) and a question, the task is to find the answer to the question based on the context. The context $C = \{c_1, c_2, \ldots c_m\}$ is described as a sequence of $m$ words and the question $Q = \{q_1, q_2 \ldots q_n\}$ a sequence of $n$ words. In the extractive setting, the answer $A$ must be a span in the context. Conversational machine comprehension is a generalization of the single-turn setting: the agent needs to answer multiple, potentially inter-dependent questions in sequence. The meaning of the current question may depend on the conversation history (e.g., in Fig. 1, the third question such as 'Where?' cannot be answered in isolation). Thus, previous conversational history (i.e., question/answer pairs) is provided as an input in addition to the context and the current question.

### 2.2 MODEL ARCHITECTURE

For single-turn MC, many top-performing models share a similar architecture, consisting of four major components: (1) question encoding, (2) context encoding, (3) reasoning, and finally (4) answer prediction. Initially the word embeddings (e.g., Pennington et al., 2014; Peters et al., 2018) of question tokens $Q$ and context tokens $C$ are taken as input and fed into contextual integration layers, such as LSTMs (Hochreiter & Schmidhuber, 1997) or self attentions (Yu et al., 2018), to *encode* the question and context. Multiple integration layers provide contextualized representations of context, and are often inter-weaved with attention, which inject question information. The context integration layers thus produce a series of query-aware hidden vectors for each word in the context. Together, the context integration layers can be viewed as conducting implicit *reasoning* to find the answer span. The final sequence of context vectors is fed into the *answer prediction* layer to select

**Context:**
<Author went to his father's funeral> After he passed away, I stayed in his apartment. I was lonely. ... <Author found a cat outside the apartment> I felt pity of him and brougut him inside. ... It has been five years since my father died. Over the years, people commented on how nice I was to save the cat. But I know we saved each other.

**Conversation Flow (over Context):**

**Time (Question turns)**

**Question:** What did he feel?

**Answer:** lonely

**Answer:** felt pity of him

**Answer:** we saved each other

Figure 2: An illustration of the conversation flow and its importance. As the current topic changes over time, the answer to the same question changes accordingly.

the start and end position of answer span. To adapt to the conversational setting, existing methods incorporate previous question/answer pairs into the current question and context encoding without modifying higher-level (reasoning and answer prediction) layers of the model.

## 3 FLOWQA

Our model aims to incorporate the conversation history more comprehensively via a conceptually simple FLOW mechanism. We first introduce the concept of FLOW (Section 3.1), propose the INTEGRATION-FLOW layers (Section 3.2), and present an end-to-end architecture for conversational machine comprehension, FLOWQA (Section 3.3).

### 3.1 CONCEPT OF FLOW

Successful conversational MC models should grasp how the conversation flows. This includes knowing the main topic currently being discussed, as well as the relevant events and facts. Figure 2 shows a simplified CoQA example where such conversation flow is crucial. As the conversation progresses, the topic being discussed changes over time. Since the conversation is about the context $C$, we consider FLOW to be **a sequence of latent representations based on the context tokens** (the middle part of Fig 2). Depending on the current topic, the answer to the same question may differ significantly. For example, when the dialog is about the author's father's funeral, the answer to the question *What did he feel?* would be *lonely*, but when the conversation topic changes to five years after the death of the author's father, the answer becomes *we saved each other*.[2]

Our model integrates both previous question/answer pairs and FLOW, the intermediate context representation from conversation history (See Fig. 1). In MC models, the intermediate reasoning process is captured in several context *integration* layers (often BiLSTMs), which locate the answer candidates in the context. Our model considers these intermediate representations, $C_i^h$, generated during the $h$-th context integration layer of the reasoning component for the $i$-th question. FLOW builds information flow from the intermediate representation $C_1^h, \ldots, C_{i-1}^h$ generated for the previous question $Q_1, \ldots, Q_{i-1}$ to the current process for answering $Q_i$, for every $h$ and $i$.

### 3.2 INTEGRATION-FLOW LAYER

A naive implementation of FLOW would pass the output hidden vectors from each integration layer during the $(i-1)$-th question turn to the corresponding integration layer for $Q_i$. This is highly unparalleled, as the contexts have to be read in order, and the question turns have to be processed sequentially. To achieve better parallelism, we alternate between them: **context integration**, processing sequentially in context, in parallel of question turns; and **flow**, processing sequentially in question turns, in parallel of context words (see Fig. 3). This architecture significantly improves efficiency during training. Below we describe the implementation of an INTEGRATION-FLOW (IF) layer, which is composed of a context integration layer and a FLOW component.

**Context Integration** We pass the current context representation $C_i^h$ for each question $i$ into a BiLSTM layer. All question $i$ ($1 \leq i \leq t$) are processed in parallel during training.

$$\hat{C}_i^h = \hat{c}_{i,1}^h, \ldots, \hat{c}_{i,m}^h = \text{BiLSTM}([C_i^h]) \tag{1}$$

---

[2]More detailed analysis on this example can be found in Appendix B.

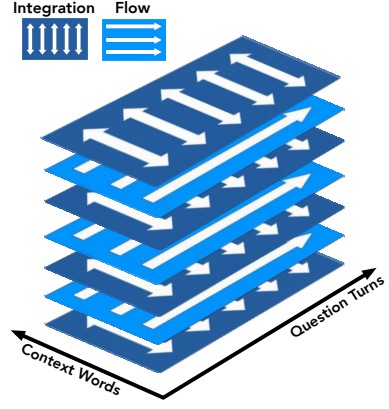

Figure 3: Alternating computational structure between context integration (RNN over context) and FLOW (RNN over question turns).

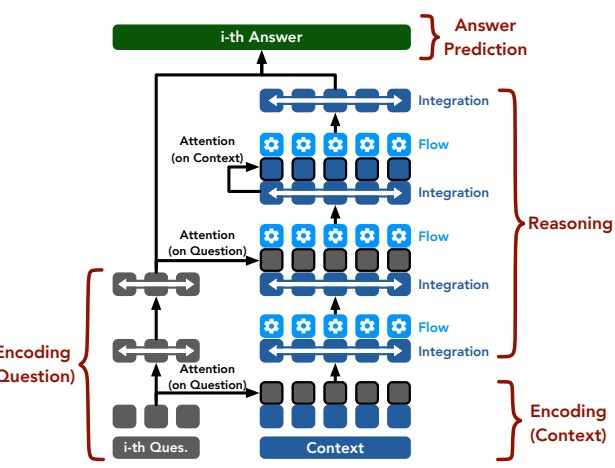

Figure 4: An illustration of the architecture for FLOWQA.

**FLOW** After the integration, we have $t$ context sequences of length $m$, one for each question. We reshape it to become $m$ sequences of length $t$, one for each context word. We then pass each sequence into a GRU[3] so the entire intermediate representation for answering the previous questions can be used when processing the current question. We only consider the forward direction since we do not know the $(i + 1)$-th question when answering the $i$-th question. All context word $j$ ($1 \leq j \leq m$) are processed in parallel.

$$f_{1,j}^{h+1}, \ldots, f_{t,j}^{h+1} = \text{GRU}(\hat{c}_{1,j}^h, \ldots, \hat{c}_{t,j}^h) \tag{2}$$

We reshape the outputs from the FLOW layer to be sequential to context tokens, and concatenate them to the output of the integration layer.

$$F_i^{h+1} = \{f_{i,1}^{h+1}, \ldots, f_{i,m}^{h+1}\} \tag{3}$$

$$\boldsymbol{C}_i^{h+1} = c_{i,1}^{h+1}, \ldots, c_{i,m}^{h+1} = [\hat{c}_{i,1}^h; f_{i,1}^{h+1}], \ldots, [\hat{c}_{i,m}^h; f_{i,m}^{h+1}] \tag{4}$$

In summary, this process takes $\boldsymbol{C}_i^h$ and generates $\boldsymbol{C}_i^{h+1}$, which will be used for further contextualization to predict the start and end answer span tokens. When FLOW is removed, the IF layer becomes a regular context integration layer and in this case, a single layer of BiLSTM.

### 3.3 FULL ARCHITECTURE OF FLOWQA

We construct our conversation MC model, FLOWQA, based on the single-turn MC structure (Sec. 2.2) with fully-aware attention (Huang et al., 2018). The full architecture is shown in Fig. 4. In this section, we describe its main components: initial encoding, reasoning and answer prediction.

#### 3.3.1 QUESTION/CONTEXT ENCODING

**Word Embedding** We embed the context into a sequence of vectors, $\boldsymbol{C} = \{c_1, \ldots, c_m\}$ with pretrained GloVe (Pennington et al., 2014), CoVE (McCann et al., 2017) and ELMo (Peters et al., 2018) embeddings. Similarly, each question at the $i$-th turn is embedded into a sequence of vectors $\boldsymbol{Q}_i = \{q_{i,1}, \ldots, q_{i,n}\}$, where $n$ is the maximum question length for all questions in the conversation.

**Attention (on Question)** Following DrQA (Chen et al., 2017), for each question, we compute attention in the word level to enhance context word embeddings with question. The generated question-specific context input representation is denoted as $C_i^0$. For completeness, a restatement of this representation can be found in Appendix C.1.

---

[3]We use GRU because it is faster and performs comparably to LSTM based on our preliminary experiments.

**Question Integration with QHierRNN**  Similar to many MC models, contextualized embeddings for the questions are obtained using multiple layers of BiLSTM (we used two layers).

$$\boldsymbol{Q}_i^1 = q_{i,1}^1, \ldots, q_{i,n}^1 = \text{BiLSTM}(\boldsymbol{Q}_i), \ \ \boldsymbol{Q}_i^2 = q_{i,1}^2, \ldots, q_{i,n}^2 = \text{BiLSTM}(\boldsymbol{Q}_i^1) \tag{5}$$

We build a pointer vector for each question to be used in the answer prediction layer by first taking a weighted sum of each word vectors in the question.

$$\tilde{q}_i = \sum_{k=1}^n \alpha_{i,k} \cdot q_{i,k}^2, \ \ \alpha_{i,k} \propto \exp(w^T q_{i,k}^2), \tag{6}$$

where $w$ is a trainable vector. We then encode question history hierarchically with LSTMs to generate history-aware question vectors (QHierRNN).

$$p_1, \ldots, p_t = \text{LSTM}(\tilde{q}_i, \ldots, \tilde{q}_t) \tag{7}$$

The final vectors, $p_1, \ldots, p_t$, will be used in the answer prediction layer.

### 3.3.2  REASONING

The reasoning component has several IF layers on top of the context encoding, inter-weaved with attention (first on question, then on context itself). We use fully-aware attention (Huang et al., 2018), which concatenates all layers of hidden vectors and uses $S(x,y) = \text{ReLU}(\mathbf{U}x)^T \mathbf{D} \, \text{ReLU}(\mathbf{U}y)$ to compute the attention score between $x, y$, where $\mathbf{U}, \mathbf{D}$ are trainable parameters and $\mathbf{D}$ is a diagonal matrix. Below we give the details of each layer (from bottom to top).

**Integration-Flow ×2**  First, we take the question-augmented context representation $C_i^0$ and pass it to two IF layers.[4]

$$\boldsymbol{C}_i^1 = \text{IF}(\boldsymbol{C}_i^0) \tag{8}$$

$$\boldsymbol{C}_i^2 = \text{IF}(\boldsymbol{C}_i^1) \tag{9}$$

**Attention (on Question)**  After contextualizing the context representation, we perform fully-aware attention on the question for each context words.

$$\hat{q}_{i,j} = \sum_{k=1}^n \alpha^{i,j,k} \cdot q_{i,k}^2, \ \ \alpha^{i,j,k} \propto \exp(S([c_i; c_{j,i}^1; c_{j,i}^2], [q_{j,k}; q_{j,k}^1; q_{j,k}^2])) \tag{10}$$

**Integration-Flow**  We concatenate the output from the previous IF layer with the attended question vector, and pass it as an input.

$$\boldsymbol{C}_i^3 = \text{IF}([c_{i,1}^2; \hat{q}_{i,1}], \ldots, [c_{i,m}^2; \hat{q}_{i,m}]) \tag{11}$$

**Attention (on Context)**  We apply fully-aware attention on the context itself (self-attention).

$$\hat{c}_{i,j} = \sum_{k=1}^m \alpha^{i,j,k} \cdot c_{i,k}^3, \ \ \alpha^{i,j,k} \propto \exp(S([c_{i,j}^1; c_{i,j}^2; c_{i,j}^3], [c_{i,k}^1; c_{i,k}^2; c_{i,k}^3])) \tag{12}$$

**Integration**  We concatenate the output from the the previous IF layer with the attention vector, and feed it to the last BiLSTM layer.

$$\boldsymbol{C}_i^4 = \text{BiLSTM}([c_{i,1}^3; \hat{c}_{i,1}], \ldots, [c_{i,m}^3; \hat{c}_{i,m}]) \tag{13}$$

### 3.3.3  ANSWER PREDICTION

We use the same answer span selection method (Wang et al., 2017; Wang & Jiang, 2017; Huang et al., 2018) to estimate the start and end probabilities $P_{i,j}^S, P_{i,j}^E$ of the $j$-th context token for the $i$-th question. Since there are unanswerable questions, we also calculate the no answer probabilities $P_i^\emptyset$ for the $i$-th question. For completeness, the equations for answer span selection is in Appendix C.1.

---

[4]We tested different numbers of IF layers and found that the performance was not improved after 2 layers.

| | Child. | Liter. | Mid-High. | News | Wiki | Reddit | Science | Overall |
|---|---|---|---|---|---|---|---|---|
| PGNet (1-ctx) | 49.0 | 43.3 | 47.5 | 47.5 | 45.1 | 38.6 | 38.1 | 44.1 |
| DrQA (1-ctx) | 46.7 | 53.9 | 54.1 | 57.8 | 59.4 | 45.0 | 51.0 | 52.6 |
| DrQA + PGNet (1-ctx) | 64.2 | 63.7 | 67.1 | 68.3 | 71.4 | 57.8 | 63.1 | 65.1 |
| BiDAF++ (3-ctx) | 66.5 | 65.7 | 70.2 | 71.6 | 72.6 | 60.8 | 67.1 | 67.8 |
| FLOWQA (1-Ans) | **73.7** | **71.6** | **76.8** | **79.0** | **80.2** | **67.8** | **76.1** | **75.0** |
| Human | 90.2 | 88.4 | 89.8 | 88.6 | 89.9 | 86.7 | 88.1 | 88.8 |

Table 1: Model and human performance (% in $F_1$ score) on the CoQA test set. ($N$-ctx) refers to using the previous $N$ QA pairs. ($N$-Ans) refers to providing the previous $N$ gold answers.

| | $F_1$ | HEQ-Q | HEQ-D |
|---|---|---|---|
| Pretrained InferSent | 20.8 | 10.0 | 0.0 |
| Logistic Regression | 33.9 | 22.2 | 0.2 |
| BiDAF++ (0-ctx) | 50.2 | 43.3 | 2.2 |
| BiDAF++ (1-ctx) | 59.0 | 53.6 | 3.4 |
| BiDAF++ (2-ctx) | 60.1 | 54.8 | 4.0 |
| BiDAF++ (3-ctx) | 59.5 | 54.5 | 4.1 |
| FLOWQA (2-Ans) | **64.1** | **59.6** | **5.8** |
| Human | 80.8 | 100 | 100 |

| | CoQA | QuAC |
|---|---|---|
| Prev. SotA (Yatskar, 2018) | 70.4 | 60.6 |
| FLOWQA (0-Ans) | 75.0 | 59.0 |
| FLOWQA (1-Ans) | **76.2** | 64.2 |
| - FLOW | 72.5 | 62.1 |
| - QHierRNN | 76.1 | 64.1 |
| - FLOW - QHierRNN | 71.5 | 61.4 |
| FLOWQA (2-Ans) | 76.0 | **64.6** |
| FLOWQA (All-Ans) | 75.3 | **64.6** |

Table 2: Model and human performance (in %) on the QuAC test set. Results of baselines are from (Choi et al., 2018).

Table 3: Ablation study: model performance on the dev. set of both datasets (in % $F_1$).

## 4 Experiments: Conversational Machine Comprehension

In this section, we evaluate FLOWQA on recently released conversational MC datasets.

**Data and Evaluation Metric** We experiment with the QuAC (Choi et al., 2018) and CoQA (Reddy et al., 2019) datasets. While both datasets follow the conversational setting (Section 2.1), QuAC asked crowdworkers to highlight answer spans from the context and CoQA asked for free text as an answer to encourage natural dialog. While this may call for a generation approach, Yatskar (2018) shows that the an extractive approach which can handle Yes/No answers has a high upper-bound – 97.8% $F_1$. Following this observation, we apply the extractive approach to CoQA. We handle the Yes/No questions by computing $P_i^Y$, $P_i^N$, the probability for answering yes and no, using the same equation as $P_i^\emptyset$ (Eq. 17), and find a span in the context for other questions.

The main evaluation metric is $F_1$, the harmonic mean of precision and recall at the word level.[5] In CoQA, we report the performance for each context domain (children's story, literature from Project Gutenberg, middle and high school English exams, news articles from CNN, Wikipedia, AI2 Science Questions, Reddit articles) and the overall performance. For QuAC, we use its original evaluation metrics: $F_1$ and Human Equivalence Score (HEQ). HEQ-Q is the accuracy of each question, where the answer is considered correct when the model's $F_1$ score is higher than the average human $F_1$ score. Similarly, HEQ-D is the accuracy of each dialog – it is considered correct if all the questions in the dialog satisfy HEQ.

**Comparison Systems** We compare FLOWQA with baseline models previously tested on CoQA and QuAC. Reddy et al. (2019) presented PGNet (Seq2Seq with copy mechanism), DrQA (Chen et al., 2017) and DrQA+PGNet (PGNet on predictions from DrQA) to address abstractive answers. To incorporate dialog history, CoQA baselines append the most recent previous question and answer to the current question.[6] Choi et al. (2018) applied BiDAF++, a strong extractive QA model to QuAC dataset. They append a feature vector encoding the turn number to the question embedding and a feature vector encoding previous $N$ answer locations to the context embeddings (denoted as $N$-ctx). Empirically, this performs better than just concatenating previous question/answer pairs. Yatskar

---

[5] As there are multiple ($N$) references, the actual score is the average of max $F_1$ against $N - 1$ references.

[6] They found concatenating question/answer pairs from the further history did not improve the performance.

(2018) applied the same model to CoQA by modifying the system to first make a Yes/No decision, and output an answer span only if Yes/No was not selected.

FLOWQA ($N$-Ans) is our model: similar to BiDAF++ ($N$-ctx), we append the binary feature vector encoding previous $N$ answer spans to the context embeddings. Here we briefly describe the ablated systems: "- FLOW" removes the flow component from IF layer (Eq. 2 in Section 3.2), "- QHIER-RNN" removes the hierarchical LSTM layers on final question vectors (Eq. 7 in Section 3.3).

**Results**    Tables 1 and 2 report model performance on CoQA and QuAC, respectively. FLOWQA yields substantial improvement over existing models on both datasets (+7.2% $F_1$ on CoQA, +4.0% $F_1$ on QuAC). The larger gain on CoQA, which contains longer dialog chains,[7] suggests that our FLOW architecture can capture long-range conversation history more effectively.

Table 3 shows the contributions of three components: (1) QHierRNN, the hierarchical LSTM layers for encoding past questions, (2) FLOW, augmenting the intermediate representation from the machine reasoning process in the conversation history, and (3) $N$-Ans, marking the gold answers to the previous $N$ questions in the context. We find that FLOW is a critical component. Removing QHier-RNN has a minor impact (0.1% on both datasets), while removing FLOW results in a substantial performance drop, with or without using QHierRNN (2-3% on QuAC, 4.1% on CoQA). Without both components, our model performs comparably to the BiDAF++ model (1.0% gain).[8] Our model exploits the entire conversation history while prior models could leverage up to three previous turns.

By comparing 0-Ans and 1-Ans on two datasets, we can see that providing gold answers is more crucial for QuAC. We hypothesize that QuAC contains more open-ended questions with multiple valid answer spans because the questioner cannot see the text. The semantics of follow-up questions may change based on the answer span selected by the teacher among many valid answer spans. Knowing the selected answer span is thus important.

We also measure the speedup of our proposed alternating parallel processing structure (Fig. 3) over the naive implementation of FLOW, where each question is processed in sequence. Based on the training time each epoch takes (i.e., time needed for passing through the data once), the speedup is 8.1x on CoQA and 4.2x on QuAC. The higher speedup on CoQA is due to the fact that CoQA has longer dialog sequences, compared to those in QuAC.

## 5   EXPERIMENTS: SEQUENTIAL INSTRUCTION UNDERSTANDING

In this section, we consider the task of understanding a sequence of natural language instructions. We reduce this problem to a conversational MC task and apply FLOWQA. Fig. 5 gives a simplified example of this task and our reduction.

**Task**    Given a sequence of instructions, where the meaning of each instruction may depend on the entire history and world state, the task is to understand the instructions and modify the *world* accordingly. More formally, given the initial world state $W_0$ and a sequence of natural language instructions $\{I_1, \ldots, I_K\}$, the model has to perform the correct sequence of actions on $W_0$, to obtain $\{W_1, \ldots, W_K\}$, the correct world states after each instruction.

**Reduction**    We reduce sequential instruction understanding to machine comprehension as follows.

- Context $C_i$: We encode the current world state $W_{i-1}$ as a sequence of tokens.
- Question $Q_i$: We simply treat each natural language instruction $I_i$ as a question.
- Answer $A_i$: We encode the world state change from $W_{i-1}$ to $W_i$ as a sequence of tokens.

At each time step $i$, the current context $C_i$ and question $Q_i$ are given to the system, which outputs the answer $A_i$. Given $A_i$, the next world state $C_{i+1}$ is automatically mapped from the reduction rules. We encode the history of instructions explicitly by concatenating preceding questions and the current one and by marking previous answers in the current context similar to $N$-Ans in conversational MC tasks. Further, we simplify FLOWQA to prevent overfitting. Appendix C.2 contains the details on

---

[7]Each QuAC dialog contains 7.2 QA pairs on average, while CoQA contains 15 pairs.

[8]On the SQuAD leaderboard, BiDAF++ outperforms the original FusionNet (Huang et al., 2018) that FLOWQA is based on.

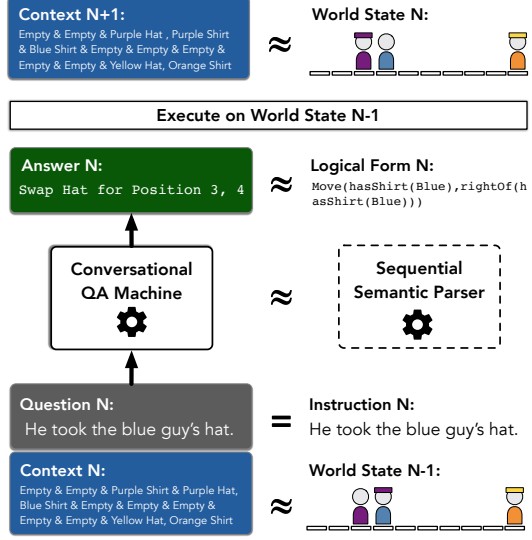

Figure 5: Illustration on reducing sequential instruction understanding to conversational MC. The corresponding units in the semantic parsing approach are shown to the right. This simplified example is from (Long et al., 2016).

|  | Sce. | Tan. | Alc. |
|---|---|---|---|
| Suhr & Artzi (2018) | 56.1 | 60.3 | 71.8 |
| FLOWQA | **64.1** | **74.4** | **84.1** |
| - FLOW | 53.0 | 67.8 | 82.0 |

Table 4: Dev accuracy (in %) after all instructions for the three domains in SCONE.

|  | Sce. | Tan. | Alc. |
|---|---|---|---|
| Long et al. (2016) | 14.7 | 27.6 | 52.3 |
| Guu et al. (2017) | 46.2 | 37.1 | 52.9 |
| Suhr & Artzi (2018) | 66.4 | 60.1 | 62.3 |
| Fried et al. (2018) | 72.7 | 69.6 | 72.0 |
| FLOWQA | **74.5** | **72.3** | **76.4** |
| - FLOW | 58.2 | 67.9 | 74.1 |

Table 5: Test accuracy (in %) after all instructions for the three domains in SCONE.

model simplification and reduction rules, i.e., mapping from the world state and state change to a sequence of token. During training, gold answers (i.e., phrases mapped from world state change after each previous instruction) are provided to the model, while at test time, predicted answers are used.

## 5.1 RESULTS

We evaluate our model on the sequential instruction understanding dataset, SCONE (Long et al., 2016), which contains three domains (SCENE, TANGRAMS, ALCHEMY). Each domain has a different environment setting (see Appendix C.2). We compare our approaches with prior works (Long et al., 2016; Guu et al., 2017), which are semantic parsers that map each instruction into a logical form, and then execute the logical form to update the world state, and (Suhr & Artzi, 2018; Fried et al., 2018), which maps each instruction into actions similar to our case. The model performance is evaluated by the correctness of the final world state after five instructions. Our learning set-up is similar to that of Fried et al. (2018), where the supervision is the change in world states (i.e., analogous to logical form), while that of Long et al. (2016) and Suhr et al. (2018) used world states as a supervision.

The development and test set results are reported in Tables 4 and 5. Even without FLOW, our model (FLOWQA-FLOW) achieves comparable results in two domains (Tangrams and Alchemy) since we still encode the history explicitly. When augmented with FLOW, our FLOWQA model gains decent improvements and outperforms the state-of-the-art models for all three domains.

## 6 RELATED WORK

Sequential question answering has been studied in the knowledge base setting (Iyyer et al., 2017; Saha et al., 2018; Talmor & Berant, 2018), often framed as a semantic parsing problem. Recent datasets (Choi et al., 2018; Reddy et al., 2019; Elgohary et al., 2018; Saeidi et al., 2018) enabled studying it in the textual setting, where the information source used to answer questions is a given article. Existing approaches attempted on these datasets are often extensions of strong single-turn models, such as BiDAF (Seo et al., 2016) and DrQA (Chen et al., 2017), with some manipulation of the input. In contrast, we propose a new architecture suitable for multi-turn MC tasks by passing the hidden model representations of preceding questions using the FLOW design.

Dialog response generation requires reasoning about the conversation history as in conversational MC. This has been studied in social chit-chats (e.g., Ritter et al., 2011; Li et al., 2017; Ghazvininejad et al., 2018) and goal-oriented dialogs (e.g., Chen et al., 2016; Bordes et al., 2017; Lewis et al., 2017). Prior work also modeled hierarchical representation of the conversation history (Park et al., 2018; Suhr & Artzi, 2018). While these tasks target reasoning with the knowledge base or exclusively on the conversation history, the main challenge in conversational MC lies in reasoning about *context* based on the conversation history, which is the main focus in our work.

## 7  CONCLUSION

We presented a novel FLOW component for conversational machine comprehension. By applying FLOW to a state-of-the-art machine comprehension model, our model encodes the conversation history more comprehensively, and thus yields better performance. When evaluated on two recently proposed conversational challenge datasets and three domains of a sequential instruction understanding task (through reduction), FLOWQA outperforms existing models.

While our approach provides a substantial performance gain, there is still room for improvement. In the future, we would like to investigate more efficient and fine-grained ways to model the conversation flow, as well as methods that enable machines to engage more active and natural conversational behaviors, such as asking clarification questions.

### ACKNOWLEDGMENTS

We would like to thank anonymous reviewers, Jonathan Berant, Po-Sen Huang and Mark Yatskar, who helped improve the draft. The second author is supported by Facebook Fellowship.

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

## A  VISUALIZATION OF THE FLOW OPERATION

Recall that the FLOW operation takes in the hidden representation generated for answering the current question, fuses into its memory, and passes it to the next question. Because the answer finding (or reasoning) process operates on top of a context/passage, this FLOW operation is a big memory operation on an $m \times d$ matrix, where $m$ is the length of the context and $d$ is the hidden size. We visualize this by computing the cosine similarity of the FLOW memory vector on the same context words for consecutive questions, and then highlight the words that have *small* cosine similarity scores, i.e., the memory that changes more significantly.

The highlighted part of the context indicates the QA model's guess on the current conversation topic and relevant information. Notice that this is *not attention*; it is instead a visualization on how the hidden memory is changing over time. The example is from CoQA (Reddy et al., 2019).

**Q1: Where did Sally go in the summer? → Q2: Did she make any friends there?**

Sally had a very exciting summer vacation . She **went to summer camp for the first** time **.** **She made friends with a girl named Tina** . They shared a **bunk** bed in their cabin . Sally 's favorite activity was walking **in** the **woods** because she enjoyed nature . Tina liked arts and crafts . Together , they made some art using leaves they found in the woods . Even after she fell in the water , Sally still enjoyed canoeing . She was sad when the camp was over , but promised to keep in touch with **her** new friend . Sally went **to the beach with** her family in the summer as well . She loves the beach . Sally collected shells **and** mailed **some** to **her friend ,** Tina , so she could make some arts and crafts with them **.** Sally liked fishing **with her brothers** , cooking on the grill **with her dad** , and swimming in the **ocean with her mother** . The summer was fun , but Sally was very excited to go **back to school . She missed her friends and teachers** . She was excited to tell them about her summer vacation .

**Q2: Did she make any friends there? → Q3: With who?**

Sally had a very exciting summer vacation . She went to summer camp for the first time . She made friends with a **girl** named **Tina** . They shared a bunk bed in their cabin . Sally 's favorite activity was walking in the woods because she enjoyed nature . Tina liked arts and crafts . Together , they made some art using leaves they found in the woods . Even after she fell in the water , Sally still enjoyed canoeing . She was sad when the camp was over , but promised to keep in touch with her new friend . Sally went to the beach **with** her **family** in the summer as well . She loves the beach . Sally collected shells and mailed some to her friend , Tina , so she could make some arts and crafts **with** them . Sally liked fishing **with** her brothers , cooking on the grill with her dad , and swimming in the ocean with her mother . The summer was fun , but Sally was very excited to go **back** to **school** . She missed her friends **and teachers** . She was excited to tell them about her summer vacation .

**Q3: With who? → Q4: What was Tina's favorite activity?**

Sally had a very exciting summer vacation . She went to summer camp for the first time . She made friends with a **girl** named **Tina** . They **shared** a bunk bed in their cabin **.** Sally **'s favorite activity was walking in** the **woods because she enjoyed nature . Tina liked arts and crafts** . Together , they made some art using leaves they found in the woods . Even after she fell in the water **,** Sally **still enjoyed canoeing** . She was sad when the camp was over , but promised to keep in touch with her new friend . Sally went to the beach with her **family** in the summer as well . She **loves** the beach . Sally collected shells and mailed some to her friend , **Tina** , so she could make some arts and **crafts** with them . Sally liked **fishing** with her brothers , **cooking** on the grill with her dad , and swimming in the ocean with her mother . The summer was **fun** , but Sally was very excited to go back to school . She missed her friends **and teachers** . She was excited to tell them about her summer vacation .

**Q4: What was Tina's favorite activity? → Q5: What was Sally's?**

Sally had a very exciting summer vacation . She went to summer camp for the first time . She made friends with a **girl** named Tina . They **shared** a bunk bed in their cabin . Sally 's favorite activity **was** walking in the woods because she enjoyed **nature** . Tina **liked arts and crafts** . Together , they made some art using leaves they found in the woods . Even after she fell in the water , Sally still

enjoyed **canoeing** . She was sad when the camp was over , but promised to keep in touch with her new friend . Sally went to the beach with her family in the summer as well . She loves the beach . Sally collected shells and mailed some to her friend , Tina , so she could make some **arts** and crafts with them . Sally liked fishing with her brothers , cooking on the grill with her dad , and swimming in the ocean with her mother . The summer was fun , but Sally was very excited to go back to school . She missed her friends and teachers . She was excited to tell them about her summer vacation .

**Q9: Had Sally been to camp before? → Q10: How did she feel when it was time to leave?**

Sally had a very **exciting** summer vacation . She went to summer camp for the first time . She made friends with a girl named Tina . They shared a bunk bed in their cabin . Sally 's favorite activity was walking in the woods because she enjoyed nature . Tina liked arts and crafts . Together , they made some art using leaves they found in the woods . Even after she fell in the water , Sally still **enjoyed** canoeing . She was **sad** when the camp was over , but promised to keep in touch with her new friend . Sally went to the beach with her family in the summer as well . She **loves** the beach . Sally collected shells and mailed some to her friend , Tina , so she could make some arts and crafts with them . Sally **liked** fishing with her brothers , cooking on the grill with her dad , and swimming in the ocean with her mother . The summer **was fun** , but Sally was very **excited** to go back to school . She missed her friends and teachers . She **was excited** to tell them about her summer vacation .

**Q16: Does she like it? → Q17: Did she do anything interesting there?** (The conversation is now talking about Sally's trip to the beach with her family)

Sally had a very exciting summer vacation . She went to summer camp for the first time . She made friends with a girl named Tina . They shared a bunk bed in their cabin . Sally 's favorite activity was walking in the woods because she enjoyed nature . Tina liked arts and crafts . Together , they made some art using leaves they found in the woods . Even after she fell in the water , Sally still enjoyed **canoeing** . She was sad when the camp was over , but promised to keep in touch with her new friend . Sally went to the beach with her family in the summer as well . She loves the beach . Sally **collected shells and** mailed **some** to her friend , Tina , so she could make some arts and crafts with them . Sally liked fishing with her brothers , **cooking on** the **grill** with her dad , **and swimming** in the ocean with her mother . The summer was fun , but Sally was very excited to go back to school . She missed her friends and teachers . She was excited to tell them about her summer vacation .

**Q18: Did she fish and cook alone? → Q19: Who did she fish and cook with?**

Sally had a very exciting summer vacation . She went to summer camp for the first time . She made friends with a girl named Tina . They shared a bunk bed in their cabin . Sally 's favorite activity was walking in the woods because she enjoyed nature . Tina liked arts and crafts . Together , they made some art using leaves they found in the woods . Even after she fell in the water , Sally still enjoyed canoeing . She was sad when the camp was over , but promised to keep in touch with her new friend . Sally went to the beach **with** her family in the summer as well . She loves the beach . Sally collected shells and mailed some to her friend , Tina , so she could make some arts and crafts with them . Sally liked fishing **with her brothers** , cooking on the grill **with her dad** , and swimming in the ocean **with her mother** . The summer was fun , but Sally was very excited to go back to school . She missed her friends and teachers . She was excited to tell them about her summer vacation .

A.1   ANALYSIS ON THE MOVEMENT OF THE MEMORY CHANGES

We found that in the first transition (i.e., from Q1 to Q2), many memory regions change significantly. This is possibly due to the fact that the FLOW operation is taking in the entire context at the start. Later on, the FLOW memory changes more dramatically at places where the current conversation is focusing on. For example, from Q4 to Q5, several places that talk about Sally's favorite activity have higher memory change, such as *was* walking in the woods, she enjoyed *nature*, and enjoyed *canoeing*. From Q16 to Q17, we can see that several memory regions on the interesting things Sally did during the trip to the beach are altered more significantly. And from Q18 to Q19, we can see that all the activities Sally had done with her family are being activated, including went to the beach *with* her family, fishing *with her brothers*, cooking on the grill *with her dad*, and swimming in the ocean *with her mother*.

Together, we can clearly see that more active memory regions correspond to what the current conversation is about, as well as to the related facts revolving around the current topic. As the topic shifts through the conversation, regions with higher memory activity move along. Hence this simple visualization provides an intuitive view on how the QA model learns to utilize the FLOW operation.

## B  EXAMPLE ANALYSIS: COMPARISON BETWEEN BiDAF++ AND FLOWQA

We present two examples, where each one of them consists of a passage that the dialog talks about, a sequence of questions and model predictions, and an analysis. For both examples, the original conversation is much longer. Here we only present a subset of the questions to demonstrate the different behaviors of BiDAF++ and FLOWQA. The examples are from CoQA (Reddy et al., 2019).

### B.1  EXAMPLE 1:

**Context:** When my father was dying, I traveled a thousand miles from home to be with him in his last days. It was far more heartbreaking than I'd expected, one of the most difficult and painful times in my life. After he passed away I stayed alone in his apartment. There were so many things to deal with. It all seemed endless. I was lonely. I hated the silence of the apartment. But one evening the silence was broken: I heard crying outside. I opened the door to find a little cat on the steps. He was thin and poor. He looked the way I felt. I brought him inside and gave him a can of fish. He ate it and then almost immediately fell sound asleep.

The next morning I checked with neighbors and learned that the cat had been abandoned by his owner who's moved out. So the little cat was there all alone, just like I was. As I walked back to the apartment, I tried to figure out what to do with him. But as soon as I opened the apartment door he came running and jumped into my arms. It was clear from that moment that he had no intention of going anywhere. I started calling him Willis, in honor of my father's best friend. From then on, things grew easier. With Willis in my lap time seemed to pass much more quickly. When the time finally came for me to return home I had to decide what to do about Willis. There was absolutely no way I would leave without him. It's now been five years since my father died. Over the years, several people have commented on how nice it was of me to rescue the cat. But I know that we rescued each other. I may have given him a home but he gave me something greater.

| Questions | BiDAF++ | FlowQA | Gold Answer |
|---|---|---|---|
| How did he feel after his dad died? | | lonely | |
| Did he expect it? | | No | |
| What did people say to him about Willis? | | how nice he was to rescue the cat | |
| Did he feel the same? | Yes | No | No |
| What did he feel? | lonely | rescued each other | rescued each other |

**Analysis:** In this example, there is a jump in the dialog, from asking about how he feel about his father's death to about Willis, a cat he rescued. The second part of the dialog resolves only around the last three sentences of the context. We can see that both BiDAF++ and FLOWQA give the correct answer to the change-of-topic question "What did people say to him about Willis". However, BiDAF++ gets confused about the follow-up questions. From BiDAF++'s answer (lonely) to the question "What did he feel?", we can speculate that BiDAF++ does not notice this shift in topic and thinks the question is still asking how the author feels about his father's death, which constitutes most of the passage. On the other hand, FLOWQA notices this topic shift, and answers correctly using the last three sentences. This is likely because our FLOW operation indicates that the part of the context on *Author's feeling for his father's death* has finished and the last part of the context on *Author's reflection about Willis* has just begun.

### B.2  EXAMPLE 2:

**Context:** TV talent show star Susan Boyle will sing for Pope Benedict XVI during his visit to Scotland next month, the Catholic Church in Scotland said Wednesday. A church spokesman said in June they were negotiating with the singing phenomenon to perform. Benedict is due to visit England and Scotland from September 16-19. Boyle will perform three times at Bellahouston Park

in Glasgow on Thursday, Sept. 16, the Scottish Catholic Media Office said. She will also sing with the 800-strong choir at the open-air Mass there. In the pre-Mass program, Boyle plans to sing the hymn "How Great Thou Art" as well as her signature song, "I Dreamed a Dream," the tune from the musical "Les Miserables" that shot her to fame in April 2009.

"To be able to sing for the pope is a great honor and something I've always dreamed of – it's indescribable," Boyle, a Catholic, said in a statement. "I think the 16th of September will stand out in my memory as something I've always wanted to do. I've always wanted to sing for His Holiness and I can't really put into words my happiness that this wish has come true at last." Boyle said her late mother was at the same Glasgow park when Pope John Paul II visited in 1982. After the final hymn at the end of the Mass, Boyle will sing a farewell song to the pope as he leaves to go to the airport for his flight to London, the church said.

| Questions | BiDAF++ | FlowQA | Gold Answer |
|---|---|---|---|
| Who is Susan Boyle? | | TV talent show star | |
| Who will she sing for? | | Pope Benedict XVI | |
| How many times will she perform? | | three times | |
| At what park? | | Bellahouston Park | |
| Where is the pope flying to? | | London | |
| What will Boyle sing? | How Great Thou Art | farewell song | farewell song |

**Analysis:** In this example, there is also a jump in the dialog at the question *Where is the pope flying to?* The dialog jumps from discussing the events itself to the ending of the event (where the pope is leaving for London). BiDAF++ fails to grasp this topic shift. Although "How Great Thou Art" is a song that Boyle will sing during the event, it is not the song Boyle will sing when pope is leaving for London. On the other hand, FLOWQA is able to capture this topic shift because the intermediate representation for answering the previous question "Where is the pope flying to?" will indicate that the dialog is revolving at around the ending of the event (i.e., the last sentence).

## C  IMPLEMENTATION DETAILS

### C.1  CONVERSATIONAL QUESTION ANSWERING

**Question-specific context input representation:** We restate how the question-specific context input representation $C_i^0$ is generated, following DrQA (Chen et al., 2017).

$$g_{i,j} = \sum_k \alpha_{i,j,k} \, \mathrm{g}_{i,k}^Q, \;\; \alpha_{i,j,k} \propto \exp(\mathrm{ReLU}(W \, \mathrm{g}_j^C)^T \, \mathrm{ReLU}(W \, \mathrm{g}_{i,k}^Q)), \tag{14}$$

where $\mathrm{g}_{i,k}^Q$ is the GloVe embedding for the $k$-th question word in the $i$-th question, and $\mathrm{g}_j^C$ is the GloVe embedding for the $j$-th context word. The final question-specific context input representation $C_i^0$ contains: (1) word embeddings, (2) a binary indicator $\mathrm{em}_{i,j}$, whether the $j$-th context word occurs in the $i$-th question, and (3) output from the attention.

$$C_i^0 = [c_1; \mathrm{em}_{i,1}; g_{i,1}], \ldots, [c_m; \mathrm{em}_{i,m}; g_{i,m}] \tag{15}$$

**Answer Span Selection Method:** We restate how the answer span selection method is performed (following (Wang et al., 2017; Wang & Jiang, 2017; Huang et al., 2018)) to estimate the start and end probabilities $P_{i,j}^S, P_{i,j}^E$ of the $j$-th context token for the $i$-th question.

$$P_{i,j}^S \propto \exp([c_{i,j}^4]^T W_S p_i), \;\; \tilde{p}_i = \mathrm{GRU}(p_i, \sum_{i,j} P_{i,j}^S c_{i,j}^4), \;\; P_{i,j}^E \propto \exp([c_{i,j}^4]^T W_E \tilde{p}_i) \tag{16}$$

To address unanswerable questions, we compute the probability of having no answer:

$$P_i^\emptyset \propto \exp\left(\left[\sum_{j=1}^m c_{i,j}^4; \max_j c_{i,j}^4\right]^T W p_i\right). \tag{17}$$

For each question $\boldsymbol{Q}_i$, we first use $P_i^\emptyset$ to predict whether it has no answer.[9] If it is answerable, we predict the span to be $j^s, j^e$ with the maximum $P_{i,j^s}^S P_{i,j^e}^E$ subject to the constraint $0 \leq j^e - j^s \leq 15$.

---

[9]The decision threshold is tuned on the development set to maximize the $F_1$ score.

**Hyper-parameter setting and additional details:** We use spaCy for tokenization. We additionally fine-tuned the GloVe embeddings of the top 1000 frequent question words. All RNN output size is set to 125, and thus the BiRNN output would be of size 250. The attention hidden size used in fully-aware attention is set to 250. During training, we use a dropout rate of $0.4$ (Srivastava et al., 2014) after the embedding layer (GloVe, CoVe and ELMo) and before applying any linear transformation. In particular, we share the dropout mask when the model parameter is shared, which is also known as variational dropout (Gal & Ghahramani, 2016). We batch the dialogs rather than individual questions. The batch size is set to one dialog for CoQA (since there can be as much as 20+ questions in each dialog), and three dialog for QuAC (since the question number is smaller). The optimizer is Adamax (Kingma & Ba, 2015) with a learning rate $\alpha = 0.002$, $\beta = (0.9, 0.999)$ and $\epsilon = 10^{-8}$. A fixed random seed is used across all experiments. All models are implemented in PyTorch (http://pytorch.org/). We use a maximum of 20 epochs, with each epoch passing through the data once. It roughly takes 10 to 20 epochs to converge.

## C.2 SEQUENTIAL INSTRUCTION UNDERSTANDING

We begin by elaborating the simplification for FLOWQA for the sequential instruction understanding task. First, we use the 100-dim GloVe embedding instead of the 300-dim GloVe and we do not use any contextualized word embedding. The GloVe embedding is fixed throughout training. Secondly, the embeddings for tokens in the context $C$ are trained from scratch since $C$ consists of synthetic tokens. Also, we remove word-level attention because the tokens in contexts and questions are very different (one is synthetic, while the other is natural language). Additionally, we remove self-attention since we do not find it helpful in this reduced QA setting (possibly due to the very short context here). We use the same hidden size for both integration LSTMs and FLOW GRUs. However, we tune the hidden size for the three domains independently, $h = 100, 75, 50$ for SCENE, TANGRAMS and ALCHEMY, respectively. We also batch by dialog and use a batch size of 8. A dropout rate of 0.3 is used and is applied before every linear transformation.

**Environment for the Three Domains**  In SCENE, each environment has ten positions with at most one person at each position. The domain covers four actions (enter, leave, move, and trade-hats) and two properties (hat color, shirt color). In TANGRAMS, the environment is a list containing at most five shapes. This domain contains three actions (add, move, swap) and one property (shape). Lastly, in ALCHEMY, each environment is seven numbered beakers and covers three actions (pour, drain, mix) dealing with two properties (color, amount).

**Reducing World State to Context**  Now, we give details on the encoding of context from the world state. In SCENE, there are ten positions. For each position, there could be a person with shirt and hat, a person with a shirt, or no person. We encode each position as two integers, one for shirt and one for hat (so the context length is ten). Both integers take the value that corresponds to being a color or being empty. In TANGRAMS, originally there are five images, but some commands could reduce the number of images or bring back removed images. Since the number of images present is no greater than five, we always have five positions available (so the context length is five). Each position consists of an integer, representing the ID of the image, and a binary feature. Every time an image is removed, we append it at the back. The binary feature is used to indicate if the image is still present or not. In ALCHEMY, there are always seven beakers. So the context length is seven. Each position consists of two numbers, the color of the liquid at the top unit and the number of units in the beaker. An embedding layer is used to turn each integer into a 10-dim vector.

**Reducing the Logical Form to Answer**  Next, we encode the change of world states (i.e., the answer) into four integers. The first integer is the type of action that is performed. The second and third integers represent the position of the context, which the action is acted upon. Finally, the fourth integer represents the additional property for the action performed. For example, in the ALCHEMY domain, $(0, i, j, 2)$ means "pour 2 units of liquid from beaker $i$ to beaker $j$", and $(1, i, i, 3)$ means "throw out 3 units of liquid in beaker $i$". The prediction of each field is viewed as a multi-class classification problem, determined by a linear layer.

