# OpenReview forum: "FlowQA: Grasping Flow in History for Conversational Machine Comprehension"
_ICLR.cc/2019/Conference_

### Official Review · AnonReviewer2 · 2018-11-02
**Strong empirical results and well written**

**Rating:** 7
**Confidence:** 4

**Review:**

In this paper, authors proposed a so-called FLOWQA for conversational question answering (CoQA). Comparing with machine reading comprehension (MRC),  CoQA includes a conversation history. Thus, FLOWQA makes use of this property of CoQA and adds an additional encoder to handle this. It also includes one classifier to handle with no-answerable questions.

Pros:
The idea is pretty straightforward which makes use of the unique property of CoQA.

Results are strong, e.g., +7.2 improvement over current state-of-the-art on the CoQA dataset.

The paper is well written.

Cons:
It is lack of detailed analysis how the conversation history affects results and what types of questions the proposed model are handled well.

Limited novelty. The model is very similar to FusionNet (Huang et al, 2018) with an extra history encoder and a no-answerable classifier.

Questions:
One of simple baseline is to treat this as a MRC task by combining the conversation history with documents. Do you have this result?

The model uses the full history. Have you tried partial history? What's the performance?

---

> ### Author Response · Authors · 2018-11-23
> **Response to Reviewer 2**
>
> Thank you for the helpful comments and clarification questions. We have added visualization for the behavior of the Flow mechanism (Appendix A) and analyzed questions where FlowQA answered correctly while previous approaches failed (Appendix B).
>
> Re: Combining the conversation history with documents
> The best performing baselines in the QuAC [1] and CoQA [2] papers indeed combine conversation history by marking previous answer locations in the evidence documents and/or concatenating questions. Effectively, these baselines reduce this problem to a regular MRC task by incorporating the conversation history in documents and questions.  These baselines’ performance is compared with that of our proposed model ( > 7.2% improvements on CoQA, and > 4.0% on QuAC).
>
> Re: Question about using partial history
> Our model incorporates the conversation history in two ways: (1) marking the previous answer locations in the evidence document as in prior baselines. (2) incorporating implicit representations generated to answer the most recent question. For the marking in the document (1), our ablation study in Table 3 shows the result for feeding in 0, 1, 2, and the full history. For incorporating implicit representations (2), our model only takes the intermediate representation generated for the most recent question (although the representation of the most recent question is based on its previous representation). The ablation study for explicit marking suggests questions often do not have a long-range dialogue dependency (most questions are related to only the preceding one or two questions).
>
> [1] Choi et al. QuAC: Question Answering in Context.
> [2] Reddy et al. CoQA: A conversational question answering challenge.

---

### Official Review · AnonReviewer3 · 2018-11-02
**Impressive experimental results but lack of clarity**

**Rating:** 6
**Confidence:** 4

**Review:**

The paper proposes a method to model the flow of context in multi-turn machine comprehension (MC) tasks. The proposed model achieves amazing improvements in the two recent conversational MC tasks as well as an instruction understanding task. I am very impressed by the improvements and the ablation test that actually shows the effectiveness of the FLOW mechanism they proposed.

However, this paper has a lack of clarity (especially, Section 3) which makes it difficult to follow and easy to lose the major contribution points of the work. I summarized the weaknesses as follows:

# lack of motivation and its validation
The paper should have more motivational questions at the beginning of why such flow information is necessary for the task. Authors already mentioned about some of it in Figure 1 and here: “such as phrases and facts in the context, for answering the previous questions, and hence provide additional clues on what the current conversation is revolving around”. However, the improvement of absolute scores in the Experiment section didn’t provide anything related to the motivation they mentioned. Have you actually found the real examples in the testing set that are correctly predicted by the FLOW model but not by the baseline? Are they actually referring to the “phrases and facts in the context”, “additional clues on what the current conversation is revolving around”? Another simple test authors can try is to show the attention between the context in a flow and question and see whether appropriate clues are actually activated given the question.

# unclear definition of “flow”
The term “flow” is actually little over-toned in my opinion. Initially, I thought that flow is a sequence of latent information in a dialog (i.e., question-answer) but it turns to be a sequence of the context of the passage. The term “flow” is more likely a sequence of latent and hierarchical movement of the information in my opinion. What is your exact definition of “flow” here? Do you believe that the proposed architecture (i.e., RNN sequence of context) appropriately take into account that? RNN sequence of the passage context actually means your attention over the passage given the question in turn, right? If yes, it shouldn’t be called a flow.

# Lack of clarity in Section 3
Different points of contributions are mixed together in Section 3 by themselves or with other techniques proposed by others. For example, the authors mention the computational efficiency of their alternating structure in Figure 2 compared to sequential implementation. However, none of the experiment validates its efficiency. If the computational efficiency is not your major point, Figure 2 and 3 are actually unnecessary but rather they should be briefly mentioned in the implementation details in the later section. Also, are Figure 2 and 3 really necessary?

Section 3.1 and 3.3.1 are indeed very difficult to parse: This is mainly because authors like to introduce a new concept of “flow” but actually, it’s nothing more than a thread of a context in dialog turns. This makes the whole points very hyped up and over-toned like proposing a new “concept”. Also, the section introduces so many new terms (“context integration”. “Flow”, “integration layers”, “conversational flow”, “integration-flow”) without clear definition and example. The name itself looks not intuitive to me, too. I highly recommend authors provide a high-level description of the “flow” mechanism at first and then describe why/how it works without any technical terms. If you can provide a single example where “flow” can help with, it would be nicer to follow it.

# Some questions on the experiment
The FLOW method seems to have much more computation than single-turn baselines (i.e., BiDAF). Any comparison on computational cost?

In Table 3, most of the improvements for QuAC come from the encoding N answer spans to the context embeddings (N-ans). Did you also compare with (Yatskar, 2018) with the same setting as N-ans?

I would be curious to see for each context representation (c), which of the feature(e.g., c, em, g) affect the improvement the most? Any ablation on this?

The major and the most contribution of the model is probably the RNN of the context representations and concatenation of the context and question at turn in Equation (4). For example, have you tested whether simple entity matching or coreference links over the question thread can help the task in some sense?

Lastly for the model design, which part of the proposed method could be general enough to other tasks? Is the proposed method task-specific so only applicable to conversational MC tasks or restricted sequential semantic parsing tasks?

---

> ### Author Response · Authors · 2018-11-23
> **Response to Reviewer 3**
>
> Thank you for detailed suggestions and feedback.
>
> Re: Question about motivation and definition of Flow
> Based on your suggestions, we have made several changes to our paper.
>
> First, we expanded the beginning of the Flow concept (section 3.1) to make the motivation clearer. We added a new figure (Figure 2) that shows a real example where existing approaches failed to answer correctly. The figure illustrates the following: depending on the current topic of the conversation, the answer to the same question can differ significantly. We hence define the conversation flow to be a sequence of latent state in the dialog, where each latent state is what the conversation (up to this point) is about. Since the conversation is based on a passage, we consider each latent state to be a block of vector representations with the same number as the passage length (e.g., the representation may store which part of the context is being discussed right now). Hence our Flow mechanism is more like a latent movement on what the relevant parts of passage currently being discussed are and not an attention over the passage.
>
> Second, to justify the motivation, we have added a visualization of the flow operation in Appendix A. Since the flow operation maintains a memory block (same size as the context length), we show where the memory update is most active (i.e., the hidden vector between each time step changes most significantly). We can see that the memory region corresponding to the current topics and events being discussed changes the most. This indicates that the model learns to use the flow operation to store information about parts of context being currently discussed. This makes the model easier to answer follow-up questions and hence leads to better performance.
>
> Third, we have added some analysis in Appendix B on dialogs where existing models failed but FlowQA succeeded. Most of them are ambiguous questions, i.e., with multiple valid answers to the question, but only one of which corresponds to the current conversation topic. For example, in an article about Susan Boyle singing for the pope, the question “What will Boyle sing?” can have several answers depending on what the circumstances are (in the main event, she will sing “How Great Thou Art”, but at the ending of the event, she will sing “farewell song”). Existing method sometimes gets confused and answer incorrectly.
>
> Re: Clarity in section 3
> Thank you for your suggestions for improving the clarity of the paper. Originally, we put all the detail in Section 3 for completeness. We have now moved the parts from existing approaches to Appendix C. Computational efficiency is one of our main practical concerns since the naive implementation is really slow. Below are the experimental results on this issue.
>
> Speedup over the naive implementation (in terms of time per epoch)
> CoQA: 8.1x
> QuAC: 4.2x
> The prediction performance after each epoch is the same, so the time to complete the training is proportional to this speedup. Since this result is quite succinct, originally we only mentioned in the main text. We have now added this result to the experiment section.
>
> Figure 2 and 3 visualizes how the speedup shown above is achieved and how the Flow component is integrated into an existing single-turn model.
>
> [1] Choi et al. QuAC: Question Answering in Context.
> [2] Yatskar et al. A Qualitative Comparison of CoQA, SQuAD 2.0 and QuAC.

---

> > ### Author Response · Authors · 2018-11-23
> > **Response to "Some questions on the experiments"**
> >
> > Re: Some questions on the experiments
> >
> > 1) Computational efficiency compared to single-turn MC: Without our alternating parallel processing structure, training time will be multiplied by the number of QA pairs in a dialog.  After implementing this mechanism, training FlowQA takes roughly 1.5x to 2x of the time training a single-turn model in each epoch.
> >
> > 2) Ablation on question-specific context representation: The features mentioned (em, g) are attention vectors obtained from the question. This is the first attention on the question (there are two attentions on the question, see Figure 4). If c is ablated, we are expecting the model to select an answer span from the context without seeing the context. In this case, the model would not work at all. The F1 scores for CoQA/QuAC without exact match feature (em), and attended question embedding (g) are reported below.
> >
> > FlowQA: 76.0 / 64.6
> > FlowQA (-em): 75.4 / 62.3
> > FlowQA (-g): 75.5 / 64.5
> >
> > 3) Improvements from encoding N answer spans: We are using the same setting for marking the previous N-answers as Choi et al. [1] and Yatskar et al. [2]. We provide a comparison below. The improvement was the biggest (7.2 F1) when marking no previous answer (0-Ans), as FlowQA incorporates history through using the intermediate representation while BiDAF++ had no access. The improvement is less pronounced but still significant (4.0 F1) when marking many previous answers.
> >       (FlowQA vs. BiDAF++)
> > 0-Ans: 59.0 vs. 51.8
> > 1-Ans: 64.2 vs. 59.9
> > 2-Ans: 64.6 vs. 60.6
> > All-Ans: 64.6 vs. N/A (3-Ans: 59.5)
> >
> > 4) Applying FLOW to other tasks: The Flow mechanism is essentially performing a large RNN update on a big memory state, which contains O(Nd) hidden units, N is the length of the passage/context and d is the hidden size per words. Due to the enormous hidden unit size, the big memory state can store all the details of the full passage/context and to operate on this large memory state. Because of the design of the Flow mechanism, we can operate on this enormous memory state efficiently. We believe the Flow mechanism can be useful for problems that require a large amount of memory, beyond the conversational MC and sequential semantic parsing. However, further investigation is needed to verify this claim.
> >
> > [1] Choi et al. QuAC: Question Answering in Context.
> > [2] Yatskar et al. A Qualitative Comparison of CoQA, SQuAD 2.0 and QuAC.

---

> ### Author Response · Authors · 2018-11-23
> **Response to "Some questions on the experiments"**
>
> This comment is moved to be below the main response.

---

### Official Review · AnonReviewer1 · 2018-11-03
**First model achieving nontrivial improvement on CoQA and QuAC datasets.**

**Rating:** 7
**Confidence:** 5

**Review:**

The paper presents a new model FlowQA for conversation reading comprehension. Compared with the previous work on single-turn reading comprehension, the idea in this paper differs primarily in that it alternates between the context integration and the question flow in parallel. The parallelism enables the model to be trained 5 to 10 times faster. Then this process is formulated as layers of a neural network that are further stacked multiple times. Besides, the unanswerable question is predicted with additional trainable parameters. Empirical studies confirm FlowQA works well on a bunch of datasets. For example, it achieves new state-of-the-art results on two QA datasets, i.e., CoQA and QuAC, and outperforms the best models on all domains in SCONE. Ablation studies also indicates the importance of the concept Flow.

Although the idea in the paper is straightforward (it is not difficult to derive the model based on the previous works), this work is by far the first that achieves nontrivial improvement over CoQA and QuAC. Hence I think it should be accepted.

Can you conduct ablation studies on the number of Reasoning layers (Figure 3) in FlowQA? I am quite curious if a deeper/shallower model would help.

---

> ### Author Response · Authors · 2018-11-23
> **Response to Reviewer 1**
>
> Thank you so much for your review.
>
> The ablation study on the reasoning layers can be found below (we count the number of context integration layers). The numbers below are the F1 scores for CoQA / QuAC, respectively. We found our original hyperparameter (4 layers) was the most effective one.
>
> # Integration layers = 3: 75.5 / 64.2
> # Integration layers = 4: 76.0 / 64.6 (original result)
> # Integration layers = 5: 75.3 / 64.1

---

### Public Comment · (anonymous) · 2019-01-22
**Clarification on SCONE**

This is a nice work on conversational QA. The authors compare with previous works on SCONE in Table 5. However, some of the previous models (Long et al., Guu et al., Suhr and Artzi) cited in Table 5 assumes a harder problem setting of learning with only the final word state, without access to intermediate answers (i.e., the gold answer_i for i = 1, 2, ..., N-1). It would be great if the authors could clarify this :)

---

> ### Author Response · Authors · 2019-01-24
> **Re: Clarification on SCONE**
>
> Hi!
>
> Yes, we are following their settings in both training and testing.
>
> From the paper description, they are testing on final world state without access to intermediate answers. But during training, they have access to the intermediate answers. For example, in (Suhr and Artzi), they wrote "During training, we create an example for each instruction." in Section 2, Learning.

---

### Meta-Review · Area_Chair1 · 2018-11-06
**novel modeling of context for conversational QA**

**Confidence:** 4
**Recommendation:** Accept (Poster)

**Metareview:**

Interesting and novel approach of modeling context (mainly external documents with information about the conversation content) for the conversational question answering task, demonstrating significant improvements on the newly released conversational QA datasets.
The first version of the paper was weaker on motivation and lacked a clearer presentation of the approach as mentioned by the reviewers, but the paper was updated as explained in the responses to the reviewers.
The ablation studies are useful in demonstration of the proposed FLOW approach.
A question still remains after the reviews (this was not raised by the reviewers): How does the approach perform in comparison to the state of the art for the single question and answer tasks? If each question was asked in isolation, would it still be the best?